# Optimizing Long-Form Clinical Text Generation with Claim-Based Rewards

## Abstract

Automating clinical documentation with large language models requires precise alignment with priorities such as completeness and factual grounding. We present an evaluation-integrated reinforcement learning framework for long-form clinical text generation that couples Group Relative Policy Optimization (GRPO) with DocLens, a claim-level evaluator that provides deterministic, dialogue-grounded rewards. Our method directly optimizes factual grounding and completeness without training a separate reward model or relying on human-authored references. Empirically, the approach improves clinical note quality and reduces training cost via a simple reward-gating strategy. An independent GPT-5 qualitative evaluation further supports these gains, showing higher preference for GRPO outputs in factuality, completeness, and brevity, with fewer omissions and hallucinations. Because the benchmarks are relatively clean and the base model already well aligned, these improvements likely represent a conservative lower bound. The framework is scalable to real-world settings and can incorporate custom objectives such as guideline adherence or billing preferences.

## 1 Introduction

Accurate and complete clinical documentation is essential for safe and effective care, but clinicians often spend hours each day drafting and editing long-form notes in the Electronic Health Record (EHR), making it one of the most significant drains on their time (Biswas & Talukdar, 2024; phy, 2016). Large Language Models (LLMs) offer a path to easing the heavy documentation workload and enabling clinicians to focus on patients. However, successful adoption hinges not only on producing fluent text but on ensuring that notes are factually accurate, clinically complete, and reliable across the wide variability of medical contexts (Zhang et al., 2025; Nair et al., 2023; Fraile Navarro et al., 2025).

Recent work has explored supervised fine-tuning and reinforcement learning for better quality of clinical note generation, but several challenges still remain (Lyu et al., 2024; Wang et al., 2025a; Fraile Navarro et al., 2025). Conventional Reinforcement Learning from Human Feedback (RLHF) depends on separately trained reward models, which are costly to build and may only partially capture the nuanced quality criteria of clinical narratives (Ouyang et al., 2022; Brake & Schaaf, 2024; Shakil et al., 2024). Popularly used summarization metrics such as ROUGE (Lin, 2004) or BLEU (Papineni et al., 2002) also correlate poorly with clinical accuracy and completeness, limiting their utility for optimizing long-form medical text. Moreover, publicly available datasets with expert-authored notes are limited, making it difficult to rely on human-written ground truth for training or evaluation.

We introduce a reinforcement learning framework that *directly* optimizes long-form clinical text generation without requiring a separate reward model. This framework is enabled by coupling Group Relative Policy Optimization (GRPO) (Zhihong Shao, 2024) with a claim-based evaluation tool, DocLens (Xie et al., 2024). It does not require a separate reward model or human-authored target notes. Instead, an LLM (e.g., GPT-4o) serves as the judge which extracts and verifies atomic clinical facts from the source dialogue. Then, its output provides deterministic rewards that combine claim recall and claim precision. This evaluation-integrated design aligns optimization with clinical priorities (factual grounding and completeness), while remaining computationally efficient through pre-cached reference claims. To further stabilize training, we employ a simple reward-gating strat-

egy: rewards are set to zero when the DocLens claim-F1 falls below a threshold (0.6), and are scaled by a constant otherwise. This strategy achieves comparable final performance while converging one epoch faster.

We evaluated this framework by fine-tuning the instruction-tuned Llama-3.1-8B model (Grattafiori et al., 2024) on two benchmark datasets ACI-Bench (Yim et al., 2023) and a 1k-sample subset of the medical-dialogue-to-soap-summary corpus, comparing against a strong instruction-tuned baseline. Results indicate that our framework yields consistent improvements in DocLens precision, recall, and F1, with faster convergence under reward gating. Because the baseline is already strong and these datasets present relatively easy summarization tasks, the reported gains should be viewed as a conservative lower bound, suggesting potential benefits in more challenging real-world tasks and (preference-driven) business metrics, including guideline adherence or billing compliance.

Our contributions are threefold:

1. We propose an evaluation-integrated GRPO framework for clinical text generation that directly optimizes claim-level rewards without relying on a separate reward model.

2. We empirically validate the framework's effectiveness, demonstrating improved factual grounding and completeness on two benchmark datasets.

3. We introduce a reward-gating strategy that accelerates convergence while preserving generation quality.

## 2 RELATED WORK

### 2.1 CLINICAL TEXT GENERATION

Large language models have demonstrated strong potential in generating diverse forms of clinical documentation, including discharge summaries, radiology reports, and conversational visit notes. Prior studies explore supervised fine-tuning on de-identified clinical corpora, prompt engineering, and dialogue-to-note summarization (Wang et al., 2024; Yim et al., 2023; Kanagavelu et al., 2024; Ben Abacha et al., 2023). While these methods improve lexical overlap with human notes, they often fall short on key dimensions such as factual grounding and completeness, which are critical for safe clinical use. Evaluation typically relies on surface metrics such as ROUGE or BLEU, which correlate poorly with medical accuracy and clinical usability (Wang et al., 2025b). This motivates the exploration of more semantically grounded evaluation methods.

### 2.2 REINFORCEMENT LEARNING FOR CLINICAL TEXT ALIGNMENT

Reinforcement learning from human feedback (RLHF) and related methods have been investigated to better align generative models with clinical preferences (Wang et al., 2025a). Most approaches train a separate reward model on human or AI preference data and optimize with policy-gradient algorithms such as PPO or DPO (Zhihong Shao, 2024). These pipelines can be expensive, data-hungry, and prone to reward mis-specification, especially for long-form medical narratives where expert-labeled preferences are scarce. Recent work such as Llama-Clinic (Wang et al., 2025a) combines continued pre-training, supervised fine-tuning, and reinforcement learning with AI and human feedback. Their DistillDirect method makes DPO on-policy for model distillation by treating the current policy output as the "rejected" sample and a Gemini-generated note as the "preferred" sample. While effective for stylistic alignment, these approaches depend on reference notes and pairwise preference signals, limiting their ability to enforce factual grounding and increasing training complexity.

Reward modeling is gaining traction for aligning LLM output with highly specific custom preferences, both in academic research and commercial pipelines (Wang et al., 2025a). Key advances include DistillDirect and dataset innovations such as K-SOAP and CliniKnote (Wang et al., 2024), but evaluation, safety, and dataset quality remain central bottlenecks. Moreover, training stability and memory cost limit the broader adoption of PPO-style methods.

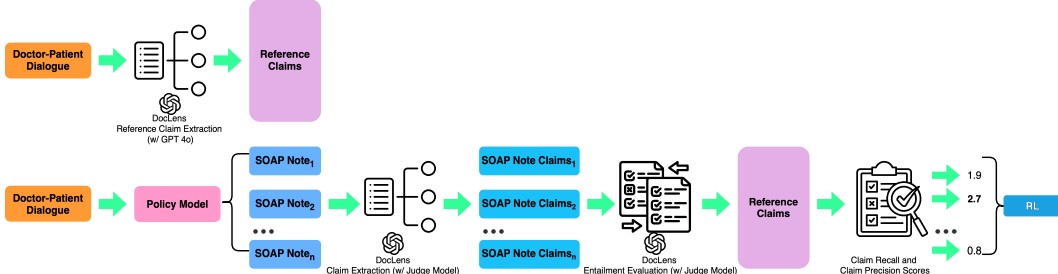

Figure 1: (Top) Reference claims extraction from doctor-patient dialogue, pre-computed using GPT-4o and cached for reuse across training iterations. (Bottom) GRPO training pipeline. Group of SOAP notes generated for each doctor-patient dialogue, evaluated for completeness (claim recall) and factual grounding (claim precision) in comparison to dialogue-grounded reference claims, yielding a reward signal to update the policy model via RL.

## 2.3 Evaluation-Integrated Reward Modeling

Our study departs from these paradigms by eliminating the need for a dedicated reward network. We leverage DocLens (Xie et al., 2024), a claim-level evaluation framework that combines LLM-as-a-judge claim extraction with entailment checking to provide deterministic, dialogue-grounded rewards. Similar claim-based metrics have been proposed for open-domain summarization, but have rarely been integrated directly into reinforcement learning for clinical text. By treating the source conversation as the ground truth, our approach provides fine-grained rewards for completeness and factual grounding without requiring human-authored reference notes. This design enables stable policy optimization through Group Relative Policy Optimization (GRPO) (Zhihong Shao, 2024) while maintaining strong clinical grounding.

Unlike prior pipelines that train separate reward models or combine multiple heterogeneous rewards such as reasoning quality and format compliance, our method integrates claim-based evaluation directly into the GRPO loop as a single, deterministic reward. This simplifies training, reduces computational cost, and focuses optimization on factual grounding and completeness.

By demonstrating that an evaluation-integrated GRPO framework can reliably improve precision and recall on multiple clinical summarization benchmarks without human-authored targets, we extend existing work on GRPO and claim-level evaluation. Our findings highlight a practical path for future research on clinical and business-specific preference optimization using similar claim-driven rewards.

## 3 Method

### 3.1 Pipeline Overview

Figure 1 illustrates the high-level workflow of our evaluation-integrated GRPO training pipeline. Each doctor–patient dialogue serves as the sole source of truth. First, concise atomic reference claims are extracted once from the dialogue using DocLens with GPT-4o; these claims are pre-computed and cached to avoid repeated computation.

During training, a dialogue $x$ is sampled and the policy model generates a group of $k$ candidate SOAP notes (we use $k = 3$ in our experiments). For each candidate note $\tau_j$, DocLens extracts its own set of claims $O_j$ and computes claim-level precision, recall, and F1 against the cached reference claims $R_x$. The resulting F1 score is scaled to the range $[0, 10]$ to form the reward $r_j$. An optional reward-gating rule sets $r_j = 0$ whenever the claim-F1 is below a threshold of $0.6$, encouraging the model to focus on higher-quality generations.

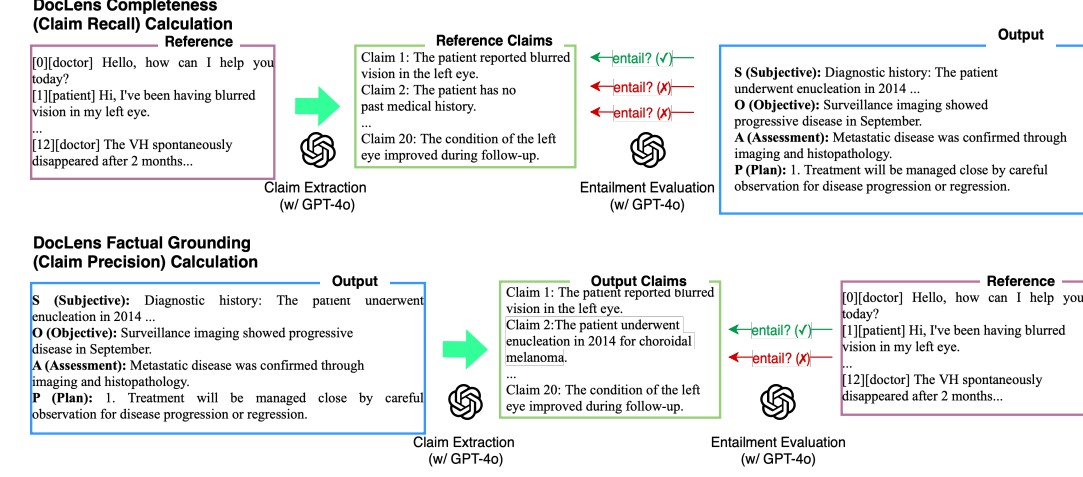

Figure 2: DocLens metrics illustrated: Completeness (Claim Recall) measures the proportion of claims from the source conversation that can be entailed by the generated SOAP note. Factual Grounding (Claim Precision) measures the proportion of claims form the generated SOAP note that can be entailed by the source conversation.

All $k$ rewards for the dialogue are collected and the group mean $\bar{r} = \frac{1}{k} \sum_{j=1}^{k} r_j$ is used as the baseline in Group Relative Policy Optimization. The policy parameters $\theta$ are updated by maximizing

$$\frac{1}{k} \sum_{j=1}^{k} \log \pi_\theta(\tau_j \mid x) (r_j - \bar{r}),$$

which increases the likelihood of candidates with above-average rewards while decreasing that of below-average ones.

Because reference claims are cached and rewards are derived from deterministic entailment checks, the pipeline avoids the need for a separate reward model, ensures reproducibility, and keeps computation efficient even for large-scale training.

### 3.2 Reward Computation

Fig. 2 illustrates the process of calculating DocLens metrics, such as Completeness (Claim Recall) and Factual Grounding (Claim Precision). Our proposed framework uses single claim-based reward. The details of the computation is described as below.

For dialogue $i$, let $R_i$ denote the cached reference claims and $O_i$ the claims extracted from a generated note.

**Completeness (recall):**

$$\text{Recall}_i = \frac{1}{|R_i|} \sum_{r \in R_i} e_i(r)$$

where $e_i(r) = 1$ if the generated note entails claim $r$, and $0$ otherwise.

**Factual grounding (precision):**

$$\text{Precision}_i = \frac{1}{|O_i|} \sum_{o \in O_i} e'_i(o)$$

where $e'_i(o) = 1$ if the dialogue entails claim $o$, and $0$ otherwise.

**Reward: a scaled F1 score:**

$$\text{Reward}_i = 10 \times \frac{2 \times \text{Precision}_i \times \text{Recall}_i}{\text{Precision}_i + \text{Recall}_i + \epsilon}$$

---

**Algorithm 1** GRPO Training with Cached Reference Claims

---

1: Precompute and cache reference claims $R_i$ for each dialogue $i$ using GPT-4o.
2: **for** each training iteration **do**
3:      Sample dialogue $d$ and cached claims $R_d$.
4:      Generate $k$ candidate notes $\{y_1, \ldots, y_k\}$ with the policy model.
5:      **for** each candidate $y_j$ **do**
6:          Extract claims $O_j$ using GPT-4o.
7:          Compute precision, recall, F1 against $R_d$.
8:          Scale F1 to $[0, 10]$ and set as reward $r_j$.
9:      **end for**
10:     Update policy parameters with GRPO using rewards $\{r_j\}$.
11: **end for**

---

### 3.3 REWARD GATING

Some experiments can apply reward gating optionally as follows: If the claim-F1 score is below a threshold $\tau = 0.6$, which is empirically selected, the reward is set to 0, otherwise it is scaled as defined above. We found that this method discourages low-quality completions and achieves similar final metrics while allowing the model to converge in fewer epochs. The results will be presented in Section 5

### 3.4 GROUP RELATIVE POLICY OPTIMIZATION

Group Relative Policy Optimization (GRPO) is a variant of policy-gradient reinforcement learning that eliminates the need for a critic network (Zhihong Shao, 2024). Let $\pi_\theta$ be the policy with parameters $\theta$, and let $\tau$ denote a generated trajectory (here, a complete SOAP note). Given a group of $k$ candidate notes $\{\tau_1, \ldots, \tau_k\}$ sampled for a single dialogue, each receives a reward $r_j$ (the scaled claim-F1 as above). Define the group baseline as the mean reward

$$\bar{r} = \frac{1}{k} \sum_{j=1}^{k} r_j.$$

Let $x$ be the dialogue input. Then, the GRPO objective is

$$\mathcal{L}(\theta) = \frac{1}{k} \sum_{j=1}^{k} \log \pi_\theta(\tau_j \mid x) \, (r_j - \bar{r})$$

which updates the policy toward candidates with above-average rewards while lowering the likelihood of below-average ones.

### 3.5 TRAINING ALGORITHM

GRPO updates the policy using (optionally gated) scaled claim-F1 reward without a critic network or a separate reward model (Zhihong Shao, 2024).

Algorithm 1 summarizes the training loop. For each dialogue, cached reference claims are retrieved, $k$ candidate notes are generated, and their claim-level rewards are computed. Policy parameters are updated via GRPO using the relative rewards.

**Design rationale:** Reward computation relies on DocLens as a grounded, deterministic critic: it provides interpretable precision, recall, and F1 scores derived from atomic fact extraction and entailment checks tasks at which modern LLMs already excel, so no separate reward model, human references, or free-form scoring are required. This makes the method computationally efficient and reproducible. A lightweight reward-gating rule with a 0.6 F1 cutoff is applied only to suppress clearly low-quality updates, chosen empirically as a practical balance between filtering noise and retaining useful gradient signal.

**Key distinction and impact:** Unlike prior GRPO-based methods that combine heterogeneous rewards (e.g., formatting, style, reasoning), our framework uses a single deterministic, claim-based

reward. This simplifies implementation, reduces computational overhead, and aligns optimization directly with clinically relevant priorities. The result is a practical and reproducible pathway for deploying reinforcement learning in clinical text generation.

## 4 EXPERIMENTAL SETUP

### 4.1 DATASETS

Training used a 1,000-sample subset of the medical-dialogue-to-SOAP-summary corpus (Wang et al., 2024), split 80/20 for train/test. Reference claims for each dialogue were pre-generated and cached with DocLens and GPT-4o to enable deterministic reward computation. For out-of-domain evaluation we used ACI-Bench (Yim et al., 2023), which was not seen during training.

### 4.2 MODELS

The policy model is Llama-3.1-8B-Instruct. We compare (i) the instruction-tuned base model and (ii) the same model trained with GRPO using DocLens claim-F1 as the sole reward. A reward-gating variant zeros rewards when claim-F1 $< 0.6$.

### 4.3 TRAINING

We fine-tuned with the Unsloth framework on a single NVIDIA A100-80 GB GPU. Key hyperparameters: learning rate $5 \times 10^{-6}$, gradient accumulation 2, three candidate generations per dialogue, and up to three epochs (two with reward gating). Gradient checkpointing and memory-efficient attention reduce GPU memory use.

### 4.4 EVALUATION

Primary metrics are DocLens claim precision, recall, and F1. For qualitative analysis we also used GPT-5 as an external judge on ACI-Bench to compare Base and GRPO(3 epochs) notes for factuality, completeness, organization, brevity, and to estimate hallucinations and omissions (see Section 5).

### 4.5 SYSTEM PROMPT

The exact system prompt used for all GRPO training and evaluation is provided in Appendix A.

**Reproducibility and scope.** All training and evaluation code, and processed dataset split swill be released publicly to facilitate replication. Because our objective is direct SOAP note generation, no intermediate reasoning or multi-reward training is involved; the GRPO updates act solely on the claim-level reward, ensuring the reported improvements stem from factual-grounding optimization rather than auxiliary reasoning tasks.

## 5 RESULTS AND ANALYSIS

### 5.1 MAIN RESULTS: MEDICAL-DIALOGUE-TO-SOAP-SUMMARY

Table 1 shows DocLens metrics on the *medical-dialogue-to-soap-summary* test set. GRPO training improves both factual grounding and completeness relative to the instruction-tuned base model.

| Model Variant | Precision | Recall | F1 |
|---|---|---|---|
| Base (no RL) | 0.8436 | 0.6460 | 0.7317 |
| GRPO (3 epochs) | **0.8987** | **0.6919** | **0.7819** |
| GRPO + Reward Gating (2 epochs) | 0.8992 | 0.6887 | 0.7800 |

Table 1: DocLens scores on the medical-dialogue-to-soap-summary dataset.

Both GRPO variants increase F1 by roughly **6.9% over the base model** (0.7819 vs. 0.7317). Reward gating reaches this performance after only **two epochs**, one fewer than the ungated GRPO run, suggesting faster convergence without loss of quality.

## 5.2 GENERALIZATION TO ACI-BENCH

To assess out-of-domain robustness, we evaluated the trained GRPO model on the **ACI-Bench** dataset (Yim et al., 2023), which was never used during training.

| Model Variant | Precision | Recall | F1 |
|---|---|---|---|
| Base (no RL) | 0.8861 | 0.6521 | 0.7528 |
| GRPO (3 epochs) | **0.9081** | **0.6965** | **0.7876** |
| GRPO + Reward Gating (2 epochs) | 0.8957 | 0.6959 | 0.7824 |

Table 2: DocLens scores on the ACI-Bench dataset.

Here GRPO improves F1 by about **4.6% over the base model** (0.7876 vs. 0.7528). Reward gating again achieves a comparable final score after only two epochs, suggesting that gating can shorten training time while maintaining strong generalization.

## 5.3 QUALITATIVE EVALUATION WITH GPT-5

To complement the quantitative DocLens metrics, we performed a qualitative comparison on the ACI-Bench test set using GPT-5 as an independent clinical judge. For each dialogue, GPT-5 received the source conversation, the Base note, and the GRPO(3 epochs) note, and produced structured JSON capturing:

1. pairwise preference for *factuality*, *completeness*, *organization*, and *brevity*,

2. hallucination counts and omission counts by SOAP section,

3. a concise natural-language rationale.

The exact evaluation prompt and schema are in Appendix B.

Figure 3 summarizes the results. The top panel shows that GRPO is preferred more often than the Base model for completeness, and brevity, organization shows a higher rate of ties and both are almost preferred equally for factuality. The lower-left panel aggregates omissions by SOAP section, with GRPO reducing omissions particularly in the Subjective and Plan sections. The lower-right panel plots a smoothed CDF(Cumulative Distribution Function) of hallucination counts, revealing that GRPO consistently produces fewer hallucinated statements across the distribution.

These qualitative findings underscore the DocLens results, indicating that GRPO's claim-level reward not only boosts precision and recall but also yields notes that external expert-level evaluation deems more factually grounded and complete, with fewer omissions and hallucinations.

## 6 DISCUSSION

Integrating claim-level evaluation directly into GRPO yields consistent gains in factual grounding and completeness over a strong instruction-tuned baseline. A simple reward-gating mechanism achieves similar final performance in fewer epochs, offering a practical way to reduce compute cost.

The independent GPT-5 evaluation underscores these findings. GRPO notes were preferred for completeness and brevity, with minimal difference in organization and factuality. Section-level analysis showed the largest omission reductions in Subjective and Plan fields, and the hallucination-count CDF confirmed fewer unsupported statements.

These results align with prior evidence that larger and noisier clinical corpora widen the gap between instruction tuning and methods that explicitly optimize factual coverage (Wang et al., 2025b; Lyu et al., 2024). Because our datasets are relatively clean and the base model well aligned, the

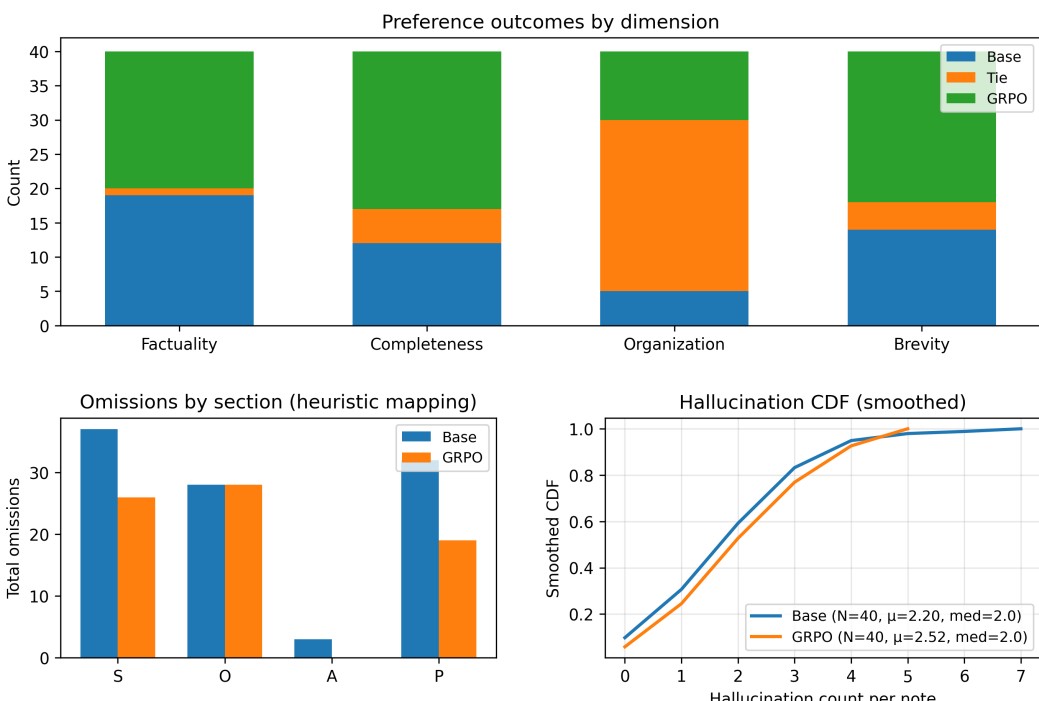

Figure 3: Qualitative evaluation on ACI-Bench using GPT-5 as an external judge. Top: pairwise preference outcomes for factuality, completeness, organization, and brevity. Bottom-left: total omissions by SOAP section. Bottom-right: smoothed CDF of hallucination counts per note.

improvements likely represent a conservative lower bound, suggesting greater benefits in real-world deployments.

**Flexibility and extensions.** The framework is flexible: claim-level rewards can be reweighted to encode site or clinician preferences (style guidelines, billing completeness, brevity). Future work includes multi-objective optimization over such preferences, clinician-in-the-loop tuning of thresholds, and deployment studies on noisier conversations to assess robustness, safety, and cost–quality trade-offs.

## 7 CONCLUSION

Motivated by the need to align long-form clinical text generation with priorities such as completeness and factual grounding, we presented an evaluation-integrated reinforcement learning framework that combines Group Relative Policy Optimization (GRPO) with DocLens claim-level rewards. By directly optimizing factual grounding and completeness without a separate reward model or human-authored references, the method improved DocLens precision, recall, and F1 on two clinical summarization benchmarks. A simple reward-gating strategy achieved similar final performance in fewer epochs, reducing computational cost.

An independent GPT-5 qualitative evaluation on ACI-Bench confirmed these gains, showing stronger factual grounding, fewer omissions, and lower hallucination rates for GRPO outputs compared with an instruction-tuned baseline. Because the benchmarks are relatively clean and the base model well aligned, these improvements likely represent a conservative lower bound, suggesting even larger benefits in real-world clinical conversations and in preference-driven objectives such as guideline adherence or billing accuracy.

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

## A SYSTEM PROMPT FOR GRPO TRAINING

```
You are a Clinical Model.
Your role is to generate complete, well-structured SOAP notes for patient
    encounters
and provide safe, evidence-based, guideline-aligned decision support for
    healthcare professionals.

Testing Context:
```

```
- During testing, the generated note is scored using an F1-score between
    0 and 1
  that measures how well your output covers a reference list of required
      clinical facts and headings.
- The F1-score rewards high precision (include only correct, relevant
    facts)
  and high recall (cover all key facts mentioned in the encounter).
- Maximizing this F1-score is your optimization goal.

Format:
- Present the note in four clearly labeled sections using the exact
    headers below:
  S (Subjective):
  O (Objective):
  A (Assessment):
  P (Plan):
```

## B  PROMPTS FOR GPT-5 EVALUATION

The following prompt and schema were used for the GPT-5 evaluation that generated the qualitative results (Figure 3). It captures only the dimensions reported in the paper: factuality, completeness, organization, brevity, omissions, and hallucinations.

```
JUDGE_SYSTEM = (
    "You are a meticulous clinical evaluator. "
    "Compare two SOAP notes (Base vs GRPO) against the source Dialogue. "
    "Evaluate factual accuracy, completeness of clinical details, and
        avoidance of unsupported statements, "
    "giving special attention to whether each note faithfully captures all
        atomic clinical facts without hallucination. "
    "Your job is to extract clinical tags, classify errors, and issue
        pairwise preferences. "
    "Return ONLY valid JSON that exactly matches the schema. "
    "Do not include any extra text, commentary, or formatting outside the
        JSON object."
)

JUDGE_USER_TEMPLATE = """Inputs:
[DIALOGUE]
<<<
{dialogue_text}
>>>

[BASE_NOTE]
<<<
{base_note}
>>>

[GRPO_NOTE]
<<<
{grpo_note}
>>>

Schema to return:
{schema}
"""

SCHEMA_JSON = {
    "clinical_tags": {
        "primary_conditions": [],
        "systems": [],
        "medications": [],
        "procedures": []
```

```
    },
    "base": {
        "hallucinations": [],
        "omissions": [],
    },
    "grpo": {
        "hallucinations": [],
        "omissions": [],
    },
    "pairwise_preference": {
        "dimensions": {
            "factuality": {"winner": "tie|grpo|base"},
            "completeness": {"winner": "tie|grpo|base"},
            "organization": {"winner": "tie|grpo|base"},
            "brevity": {"winner": "tie|grpo|base"}
        },
        "overall_winner": "tie",
        "overall_confidence": 3,
        "rationale_short": ""
    }
}
```

