# OpenReview forum: "Optimizing Long-Form Clinical Text Generation with Claim-Based Rewards"
_ICLR.cc/2026/Conference — Submitted to ICLR 2026_

### Official Review · Reviewer_e6DB · 2025-10-27

**Soundness:** 2
**Presentation:** 3
**Contribution:** 2
**Rating:** 2
**Confidence:** 4

**Summary:**

This paper presents an evaluation-integrated reinforcement learning framework for clinical SOAP note generation that combines Group Relative Policy Optimization (GRPO) with DocLens, a claim-based evaluation tool. The key contribution is using DocLens directly as the reward signal, eliminating the need for training a separate reward model as in traditional RL with human feedback approaches. Reference claims are extracted using GPT-4o and used to optimize the policy model to maximize claim-level precision and recall through GRPO. A reward gating strategy is also introduced to accelerate convergence. Empirical validation of the dataset is performed on a medical-dialogue-to-SOAP dataset (in-domain) and an ACI-Bench dataset (out-of-domain) with qualitative evaluation using LLM as a judge.

**Strengths:**

* A RL framework that does not require the traditional costly reward model training and instead uses the DocLens evaluation to serve as a deterministic reward signal.
* Performance evaluated on an in-domain dataset and an out-of-domain evaluation (ACI-Bench) to assess generalization.
* Honest empirical assessment of the performance improvement on two different datasets with the notion that performance is already quite good and thus expectation of larger gains is impractical.
* Well-structured paper that was clear to understand and motivates the problem well.

**Weaknesses:**

* There are some experimental setup issues that limit the impact of the results.
  - The paper only compares a single instruction-tuned base model (Llama-3.1-8B-Instruct) against the GRPO and GRPO + reward gating without comparison against other LLM backbones. It's unclear if the framework generalizes broadly to other LLMs (or even potentially some of the medical-oriented variants like PMC-LLama, MedGemma, etc.)
  - There are no comparisons with other RL methods using PPO, DPO, or DistillDirect (mentioned in related work).
  - The paper uses DocLens both as a training objective _and_ the primary evaluation metric. This means the optimized model should perform better, but it's not clear if it means clinical utility is improved. For example, how is hallucination calculated. Is it when there's a medical claim that exists in the note but wasn't in the extracted one using GPT-4o?
  - There are no ablation studies for some of the key choices in the model. For example, why is the reward gating using a threshold of 0.6? How much does that impact convergence? What about group size of 3?
* The experiments rely on LLM as a judge and as the oracle. There are no discussion as to whether there is error or biases from GPT-4o that may affect the final model (there is no analysis of GPT-4o's reliability for medical claim verification in fact). Similarly, a single LLM is used as a judge for the ACI Bench. Why isn't there any analysis to compare with human expert judgements? At the very least, a different LLM backbone should've been used for evaluation (Prometheus-Eval is an example of a dedicated LLM-As-A-Judge framework that might also potentially make sense).
 * DocLens is used as a crucial component of the framework but doesn't validate the quality (e.g., is the claim extraction from GPT-4o consistent as defined in the deterministic rewards, does it match with human expert assessments of note quality as the original DocLens paper does not benchmark on this dataset)?
* There is no training convergence comparison even though it is one of the major claims of the benefts of the model. The proposed model uses only 3 epochs (2 with reward gating) but are not benchmarked against other RL approaches? Is this due to the deterministic nature or that it overfits?

Minor
* It's unclear why there is a need to repeatedly state results are a conservative lower bound. Is this because the improvement is small?
* Please introduce acronyms once and use them consistently throughout the test (e.g., GRPO is introduced twice, but PPO and DPO aren't ever introduced).

**Questions:**

1. Why were no other RL methods included in the comparison?
2. Does your framework generalize to other LLM backbones (both generalist and medical-domain specific)?
3. Does optimizing DocLens F1 improve clinical utility or does it teach the model to match GPT-4o's?
4. How does convergence compare with other RL methods and what evidence do you have that your model has converged within 2 or 3 epochs?
5. Is the 0.6 threshold used for reward gating optimal? What about the sensitivity to the group size?
6. Do human experts agree on the evaluation(e.g., is GRPO better than base notes)?

---

### Official Review · Reviewer_sot1 · 2025-10-30

**Soundness:** 2
**Presentation:** 1
**Contribution:** 2
**Rating:** 2
**Confidence:** 4

**Summary:**

The paper proposes an evaluation-integrated RL pipeline for clinical note generation that combines GRPO with DocLens as tool that extracts claim sets from the source dialogue and from generated SOAP notes to calculate reward (F1-over claims). The results show that on the med-dialogue-to-SOAP test subset and ACI-Bench with Llama-3.1-8B-Instruct, the approach improves F1 performance against instruction-tuned base model baseline. The reward gating variant also proposed in the work achives the same performance more efficiently by using less number of epochs. The paper also provides with GPT-5 based qualitative eval reports to demonstrate effectiveness of the approach over hallucination rate, completeness, organisation, brevity.

**Strengths:**

- Simple pipeline: Uses GRPO with a single claim-based reward without the need for a separate reward model to train is efficient and interesting for the specific clinical generation task.
- The results show small yet steady improvements on both in-domain and ACI-Bench (out-of-domain).
- The qualitative results discussed in the paper provides interpretability to effectiveness of the proposed approach in tackling hallucination and improving factuality, completeness and brevity.

**Weaknesses:**

- Weak baseline for claiming superior performance: Missing comparisons to (1) rerank-only with DocLens (no RL), (2) standard PPO/DPO with reward model, (3) supervised training on extracted claims. Also the baseline in the paper is the Llama-3.1-8B-Instruct and not Llama-3.1-8B-Instruct with SFT on this data, which should be the first and most important comparison against the proposed GRPO. It is obvious that the base model will perform poorly against a model trained on this dataset. The results are only interesting if they improve over the SFT model trained on the same dataset.
- Claiming the DocLens based rewards are “Deterministic rewards” is misleading: DocLens relies on an LLM (GPT-4o) for claim extraction/entailment. Without strict decoding control, rewards are not strictly deterministic.
- Small scale experiments to support reliability of the claims in larger scale training scenarios. It is unsure if the proposed approach can be still superior to SFT models on similar data when trained for longer. Although I acknowledge the low resource nature of clinical domain, right now the the training dataset size ~800 samples for 2/3 epochs, which is too small. (Although I can understand if the authors had resource constraints)
- Lack ablations: No sensitivity study for choosing the 0.6 gate threshold, reward scaling to (0-10] (why not 0-1?), or group size k [L262-238].
- The discussion on qualitative evaluations using GPT-5E lacks reliability and confidence reporting. Although GPT-5 is used as an annotator, the resulting error analysis is hard to trust without inter-annotator agreement or multiple human/LLM annotators.

**Questions:**

- " What is SOAP?" should be defined explicitly before it is first mentioned in L59-60.
- Why was Base model w SFT not considered as a baseline?
- Why were experiments conducted on small setting? any results on how the claims stand with longer training?
- How the models perform when only precision or recall is used as reward?
- Why scale rewards to [0,10] instead of [0,1]? Did you observe stability differences with GRPO?

---

### Official Review · Reviewer_8ZfN · 2025-11-02

**Soundness:** 1
**Presentation:** 2
**Contribution:** 2
**Rating:** 0
**Confidence:** 3

**Summary:**

The authors seek to demonstrate a novel evaluation-integrated reinforcement learning for clinical text generation in documentation contexts. They include the DocLens framework in a GRPO loop, and find improved overall performance on ACI-Bench. They complete the paper by offering qualitative evaluation performed via an instance of GPT-5.

**Strengths:**

- The authors offer a reasonable summary of the existing literature regarding document summarization.
	- Their use of the atomic claims framework as applied to document summarization is reasonably novel - particularly their described integration directly into the reward loop. They show reasonably convincingly that this improves performance on the ACI-BENCH overall, which offers reasonable proof-of-concept assessment of this method.
	- In particular, the use of non-LLM, traditional NLP methods to show that the inclusion of LLM-as-judge in the training loop is useful is both reasonable and interesting, clearly establishing the promise of the doclens atomic claims method in this space. While this is not as robust as the use of a final human validation, and far from sufficient to rely upon, it remains useful as a directional signal. I do have very substantial concerns about the GPT-5 section (as outlined below), however, as discussed below.
	- The diagrams are reasonably clear, and the overall work is straightforward to follow.
The appendix is well-documented and clear.

**Weaknesses:**

- I strongly dislike the authors descriptions of "GPT-5 as an external judge" in this paper, without further human validation. Use of LLM-as-judge in this way is inappropriate without reference to whether it performs this task equivalently to a human. This is especially true for complex, clinically relevant text such as that in the notes that the authors are seeking to evaluate. If the errors are so subtle as to be invisible to the first LLM, what is the guarantee that they will be visible to the second? It might be countered that GPT-5 is substantially larger and more powerful overall than the LLAMA model tested, but this then would obviously heavily limit the scalability of this approach as extended to the frontier.
	- To further extend the point above, while the DocLens framework was itself validated against clinical context, this human validation remains a core metric for relying on it in any capacity (see https://ai.nejm.org/doi/full/10.1056/AIe2500143). You cannot extend this to a prompted model without similar evaluation or at least.
	- To make claims calling this "external expert-level evaluation" is misleading to the point of absurdity here, without validation against actual experts. The inclusion of this section at all essentially leads to noise in the paper. LLMs clearly have stylistic preferences which may not align with those of human authors, the so-called "slop" phenomenon (also see https://www.nejm.org/doi/full/10.1056/NEJMp2405999). An LLM-LLM loop integrating solely LLM preferences cannot and must not become acceptable as a metric and risks leading to runaway flaws in note quality.
	- There is a clear difference between the use of independently validated systems (or those using more deterministic, non-LLM methods like ACI-bench) in conjunction, and creating novel unvalidated systems from whole cloth. Even these conjunctions require repeated validation before clinical deployment, although this is not necessarily essential at this level of conceptual evaluation.
	- I must firmly recommend rejection of this paper while this GPT-5 section remains. I defer to the AC as to whether the GRPO-->ACI-Bench improvement work (which is reasonable in its own right) is sufficient to stand as an acceptably novel contribution to the literature.

**Questions:**

1. Was any validation whatsoever performed on this GPT-5 based "qualitative assessment" system? If so, what?

---

### Official Review · Reviewer_Srmz · 2025-11-03

**Soundness:** 1
**Presentation:** 1
**Contribution:** 1
**Rating:** 2
**Confidence:** 5

**Summary:**

This paper applies a reinforcement learning framework for long-form clinical text generation to the “doctor–patient dialogue → SOAP note” task. The method uses GRPO for policy updates and employs DocLens as a claim-level reward. DocLens computes completeness (recall) and factual grounding (precision), combining them into an F1 reward that eliminates the need for a separate reward model. During training, reference claims from each dialogue are cached; for each generated note, new claims are extracted and compared with the reference to compute the reward used for GRPO updates. The authors also include a qualitative pairwise evaluation using GPT-5. Experiments focus mainly on this dialogue-to-SOAP task and report one generalization test on ACI-Bench.

**Initial Recommendation:** Reject.
The baselines and experimental design are too simplistic. The paper does not convincingly show the necessity of RL, nor that the GRPO + DocLens combination outperforms simpler training and reward schemes. The reporting and reproducibility are incomplete. If these issues are addressed during the rebuttal, I would be open to revisiting my evaluation.

**Strengths:**

* Directly integrates a reproducible claim-level metric into RL, with a clear design. The reward based on interpretable precision and recall aligns well with the task objectives.
* Avoids training a separate reward model, so it's simple and computationally efficient, as the paper emphasizes.
* The application task itself is practically meaningful: generating SOAP notes from medical dialogues is a relevant real-world clinical scenario.

**Weaknesses:**

* **Lack of motivation for using RL.** DocLens can also serve as a data filter or scorer to build datasets for SFT or DPO. The paper does not provide a fair comparison against SFT and DPO baselines under equal data and compute budgets. It is unclear why RL is necessary.
* **Overly narrow reward design.** The paper claims that using DocLens as a reward is advantageous, but does not compare against simpler reward functions, e.g., ROUGE-F1, a lightweight LLM-judge score (G-eval style), or rank-based rewards for candidates ranking during rollout. These alternatives are cheaper and could be equally effective. The design’s transferability to other long-form clinical tasks is not demonstrated.
* **Metric–evaluation coupling risk.** DocLens is both the reward and the primary evaluation metric. Since it is imperfect (no evaluation metric is pefect!), the observed improvements may stem from metric overfitting. No blinded human evaluation is provided, making the clinical significance uncertain.
* **Insufficient reporting of training cost and settings.** Missing details include candidate count (k) sensitivity, reward gating threshold ablation, token usage, runtime, and cost per example.
* **Reproducibility concerns.** The appendix contains only prompts, without additional important details for community reproduction.
* **Incomplete disclosure of LLM usage.** ICLR requires authors to disclose significant use of large language models; this information is currently missing.

**Questions:**

1. Why is RL necessary? Please provide a fair comparison under identical budgets:

   * **SFT:** Use DocLens to select high-quality instruction pairs.
   * **DPO:** Use DocLens to construct preference pairs.
   * **GRPO:** Use DocLens as the reward function.
   * Report their performance under the same model, data, candidate number (k), and training steps.
2. Add at least three lightweight alternatives: ROUGE-F1, a G-Eval/LLM-judge scalar score, and a multi-rollout rank-based reward. Compare their effectiveness and computational cost to DocLens-F1.
3. I believe that experimental results for this task, without expert evaluation, have a limited real impact. Perhaps it's just a reward hack of the DOCLENS metrics; without human evaluation and qualitative analysis/case study, it doesn't convince me.
4. Report candidate number per case, token usage, training steps, GPU memory/time, inference latency, and overall cost. Analyze the effects of candidate count (k), reward gating threshold, and claim extraction errors on final performance. Include detailed data processing steps, hyperparameter tables, random seeds, and released code/version information.
5. Add a clear LLM usage disclosure section and an ethics statement describing data privacy and compliance procedures.

---

### Meta-Review · Area_Chair_qvXc · 2026-01-03

**Summary:**

All four reviewers would like to reject the paper, with one reviewer gave 0 (strong reject).

Reviewers are concerned about evaluation metrics, weak baselines, usage of GPT-5 as a judge without analyzing its reliability, reproducibility, lack of ablation, missing experiment details, incomplete disclosure of LLM usage, etc.

**Reviewer Concerns:**

The authors did not provide any response.

**Reviewer Scores:**

The scores would remain the same since the authors did not provide any response.

---

### Decision · Program_Chairs · 2026-01-26

Reject